# Effects of Low-Level Laser Therapy on Dentin Hypersensitivity in Periodontally Compromised Patients Undergoing Orthodontic Treatment: A Randomised Controlled Trial

**DOI:** 10.3390/jcm12041419

**Published:** 2023-02-10

**Authors:** Zhiyi Shan, Chong Ren, Min Gu, Yifan Lin, Fung Hou Kumoi Mineaki Howard Sum, Colman McGrath, Lijian Jin, Chengfei Zhang, Yanqi Yang

**Affiliations:** Faculty of Dentistry, The University of Hong Kong, 34 Hospital Road, Sai Ying Pun, Hong Kong SAR, China

**Keywords:** dentin hypersensitivity, orthodontics, periodontally compromised patient, low-level laser therapy

## Abstract

Objectives: This study aimed to assess the effects of low-level laser therapy (LLLT) on dentin hypersensitivity (DH) in periodontally compromised patients undergoing orthodontic treatment. Methods: This triple-blinded randomised controlled trial included 143 teeth with DH from 23 periodontally compromised patients. Teeth on one side of the dental arch were randomly assigned to the LLLT group (LG), while those on the contralateral side were allocated to the non-LLLT group (NG). After orthodontic treatment commenced, patients’ orthodontic pain (OP) perceptions were documented in pain diaries. DH was assessed chairside by a visual analogue scale (VAS_DH_) at fifteen timepoints across the orthodontic treatment and retention. VAS_DH_ scores were compared among timepoints by the Friedman test, among patients with varying OP perceptions using the Kruskal–Wallis tests, and between the LG and NG with the Mann–Whitney U test. Results: DH generally decreased over the observation (*p* < 0.001). The VAS_DH_ scores differed among patients with varying OP perceptions at multiple timepoints (*p* < 0.05). The generalized estimating equation model showed teeth in the LG had a significantly lower VAS_DH_ score than the NG at the 3rd month of treatment (*p* = 0.011). Conclusion: LLLT could be potentially beneficial in managing DH in periodontally compromised patients undergoing orthodontic treatment.

## 1. Introduction

Patients with severe periodontal breakdown have a high risk of pathologic tooth migration due to excessive masticatory forces, typically displaying proclination of incisors and dispersed spaces among anterior teeth segments [1,2]. Nowadays, an increasing number of middle-aged populations with periodontal problems seek orthodontic care after inflammation control for better oral function and aesthetics [3]. Current evidence supports that orthodontic treatment could benefit periodontal health by facilitating occlusal force redistribution and equilibrium [4,5,6] under the prerequisite of achieving an inflammation-remitted and stabilized periodontal condition [7]. Sustaining a fully controlled periodontal inflammation is extremely important for periodontally compromised patients during and after orthodontic treatment [8]. Although guided by professional care and instructions, patients are the central executive of their daily oral hygiene and inflammation control [9,10]. Periodontally compromised patients need to self-regulate their oral hygiene intensively, achieve a stable and relatively healthy periodontal status, and guarantee a successful periodontal–orthodontic interdisciplinary treatment [11].

For periodontally compromised patients undergoing orthodontic treatment, unfavourable self-perceptions are critical as they impede oral hygiene behaviours and deteriorate periodontal conditions [12,13,14,15]. Dentin hypersensitivity (DH), characterized by a short and sharp pain in response to external stimuli, was reported among 72.5% to 98% of adults with periodontal diseases [16,17,18,19]. According to the “hydrodynamic theory”, external stimuli result in rapid shifts of the fluids within the dentinal tubules and initiate sensor nerve activation in the pulp and dentine region [20]. The high DH prevalence of periodontally compromised patients is associated with an increasing amount of exposed dentin tubules due to gingival recession, clinical attachment loss, or possible cementum removal during periodontal debridement [21]. Another prevalent unpleasant self-perception experienced by patients undergoing orthodontic treatment is orthodontic pain (OP), characterized by soreness, pressure, and tenderness [22]. OP is related to local ischemia in reaction to orthodontic forces and the activation of sensory endings in the periodontal tissues [23,24]. Previous investigations regarding the mechanism of DH and OP were separate. In the periodontal–orthodontic context, DH and OP may coexist with potential biomedical interactions, since the trigeminal nerve is the common neural pathway that connects them to the somatosensory cortex [22,25]. It is, however, unclear how DH and OP interact during periodontal–orthodontic treatment. Psychological studies suggest that people’s perceptions determine their actions, and pain could affect health behaviours [26,27,28]. For this reason, periodontally compromised patients, in particular, require therapeutic management of DH and OP in order to establish a comfortable and supportive oral hygiene environment during orthodontic treatment [15,29].

Multiple attempts have been made to alleviate DH and OP. Low-level laser therapy (LLLT) presents its strength as a novel, safe, and non-invasive approach for regulating biological activities in the dentine–pulp complex and surrounding periodontal tissues without provoking thermal effects [30]. On the one hand, previous evidence supports satisfactory desensitising effects of LLLT on DH for general populations without orthodontic intervention [31]. On the other hand, LLLT was reported to have favourable effects on OP alleviation for orthodontic patients under an optimal range of therapeutic settings [32]. For periodontally compromised patients, one clinical study also detected that LLLT (940 nm and 800 mW) alleviated OP in periodontally compromised patients and inhibited the elevation of interleukin-1β, prostaglandin E2, and substance P during the first month of active treatment [33]. Considering potential biomedical interactions and the common neural transmission pathway between OP and DH, LLLT might be a promising candidate to manage DH for periodontally compromised patients undergoing orthodontic treatment. Yet, no studies are available to evaluate the effects of LLLT on tooth DH in the periodontal–orthodontic context.

To fill in the above research gaps, this randomised controlled trial was conducted with three objectives: (1) observing DH for teeth in periodontally compromised patients during orthodontic treatment and retention, (2) analysing DH in periodontally compromised patients with varying perceptions of OP, and (3) evaluating the effects of LLLT on DH for teeth in periodontally compromised patients undergoing orthodontic treatment.

## 2. Materials and Methods

This study is a triple-blinded, two-arm randomised controlled trial with a split-mouth design. The protocol was registered on the ClinicalTrial.gov website (ID: NCT03765151). Following the Declaration of Helsinki (version 2008), ethical approval was authorized by the Institutional Review Board of the University of Hong Kong/Hospital Authority Hong Kong West Cluster (HKU/HA HKW IRB, reference number: UW 18-131). Patients were recruited from Orthodontics and Periodontics Department at Prince Philip Dental Hospital from 2016 to 2018. Written informed consent was obtained from all participants.

### 2.1. Eligibility Criteria for the Subjects

#### 2.1.1. Inclusion Criteria

Teeth in adult patients of Chinese ethnicity with treated and stabilized periodontitis who were about to undertake adjunctive orthodontic treatment [34] as a part of the occlusal therapy.Teeth in patients whose maximum contact point displacement was greater than 4 mm taking reference to the item 4d in Dental Health Component of the Index of Orthodontic Treatment Need (IOTN-DHC) [35,36].Teeth with DH perceptions in response to a chairside air-blast stimulation before orthodontic treatment started.Incisors, canines, and premolars in semi-arches adaptive to probe coverage of the LLLT device without the need for repeated lasers for each treatment session.

#### 2.1.2. Exclusion Criteria

Teeth with caries, unsatisfactory restorations, or non-carious cervical lesions close to the pulp chamber (simplified score 0 or 1 for tooth wear index [37,38]).Teeth displayed any indication of pulpitis, pulp necrosis, or acute and chronic inflammation of the periapical areas.Teeth that had been subject to trauma, surgery, or invasive periodontal treatment within the past three months.Teeth from patients who were pregnant or lactating, taking systemic medications, or using desensitising toothpaste.Teeth from patients who required comprehensive orthodontic treatment since teeth could bear heavier forces and undergo long-distance movement than adjunctive orthodontic treatment [39,40].Teeth from patients with severe craniofacial abnormalities, temporomandibular diseases, trigeminal neuralgia, or migraine could affect their subjective judgment.

### 2.2. Sample Size Calculation

Sample size determination was based on a previous study with similar LLLT settings (940 nm and 1000 mW) [41]. To detect a difference in DH between LLLT (1.36 ± 2.02) and non-LLLT (2.50 ± 2.09) groups over two weeks, 82 teeth were calculated at the 0.05 significance level to reach 80% power by the software G*Power version 3.1.9.2 [42]. Considering a 15% dropout rate, at least 123 teeth were required to recruit for this study.

### 2.3. Study Design

This randomised controlled trial was triple-blinded, following a split-mouth design. A statistician conducted the randomisation process using a computer program, and the allocation sequence was concealed in a batch of opaque envelopes. Only one designated surgery assistant, who did not participate in the study design and assessment, was allowed to check the allocations, and pre-adjust parameters of the laser device before each treatment visit. Patients, clinicians, and outcome assessors were all blinded to the allocation sequence.

### 2.4. Interventions

#### 2.4.1. Orthodontic Treatment

All patients fulfilling the above eligibility criteria were subject to adjunctive orthodontic treatment using pre-adjusted fixed appliance (0.022-inch slot system, MBT, 3M Unitek, Monrovia, CA, USA), which was provided by the same experienced operator (Y.Y.). Individual treatment plans and goals were made via discussion between orthodontists and periodontists and then finalized with patients’ approval. Accordingly, orthodontic forces were delivered to teeth for designated movement. A standardized archwire protocol at the initial treatment stage was employed considering its effect on orthodontic pain [22,35,43], specifically, with a 0.014-inch thermal nickel–titanium (NiTi) wire (G&H Orthodontics, Franklin, IN, USA) for the first two months, followed by a 0.016-inch thermal NiTi wire (G&H Orthodontics, Franklin, IN, USA) for one month. By then, most teeth had achieved a preliminary alignment. Subsequently, 0.018-inch, 0.017 × 0.025-inch NiTi archwires (G&H Orthodontics, Franklin, IN, USA), and 0.017 × 0.025-inch stainless steel archwires (G&H Orthodontics, Franklin, IN, USA) were sequentially administered to the patients depending on individual circumstances. After anticipated treatment outcomes were achieved, fixed appliances were removed, and orthodontic treatment entered the retention phase with immediate delivery of polyvinyl chloride retainers and fixed-lingual retainers (0.0215-inch multistranded wire) attached to the anterior teeth.

#### 2.4.2. Low-Level Laser Therapy (LLLT)

Teeth on one side of the dental arch were randomly allocated into the LLLT group and received repeated 940 nm wavelength diode laser (EZlase; Biolase Technology Inc., Irvine, CA, USA) by a quadrant-size probe (beam size: 2.8 cm^2^). Teeth on the contralateral side were allocated into the non-LLLT and subjected to pseudo-laser irradiation with an identical appearance and sound. The output power on the test side was confirmed as 800 mW by a power metre (OPHIR Nova II Power Metre, Ophir-Spiricon, LLC, Logan, UT, USA), while that for the contralateral side was 0 mW. During the irradiation, the quadrant-size probe was firstly placed 1 mm above the buccal cervical areas of the central incisor to the second premolar for 30 s (8.6 J/cm^2^ energy density for the test side). Then the probe was shifted apically and placed 1 mm above the gingival mucosa covering root regions for another 30 s with the same parameter setting. LLLT was administered repeatedly during the active treatment and retention stage following the schedule shown in Figure 1.

### 2.5. Outcome Assessment

Periodontitis stage and grade for each patient before orthodontic treatment were evaluated by one calibrated examiner (Z.S.) based on the new classification scheme for periodontal diseases [44,45].

DH for all included teeth was determined by patients’ subjective ratings on one 100-mm visual analogue scale (VAS_DH_, 0 = no sensitivity, 100 = worst possible sensitivity). Air-blast stimulation was delivered by the same assessor (Y.Y.) using the same triple syringe in one dental chair with a 5 mm distance above the cervical area to tooth labial/buccal surfaces. Participants were instructed to place a mark on the VAS_DH_ ruler immediately after each stimulation. The scores were recorded at the baseline (T_pre_), immediately after orthodontic force activated (T_immd_), and at 1 week (T_1w_), 1 month (T_1m_), 3 months (T_3m_), 6 months (T_6m_), and 12 months (T_12m_) during active orthodontic treatment. VAS_DH_ scores for the included teeth were obtained immediately after debonding (T_deb_), and at 1 week (TR_1w_), 2 weeks (TR_2w_), 3 weeks (TR_3w_), 1 month (TR_1m_), 3 months (TR_3m_), 6 months (TR_6m_), and 12 months (TR_12m_) during orthodontic retention stage. The fifteen observation timepoints are illustrated in Figure 1 with red vertical lines.

After appliance fixation, every patient was provided with a standardized pain diary to document their initial OP perceptions on the 100 mm VAS_OP_ scale (0 = no pain, 100 = worst possible pain) for seven days. The duration of OP was determined by counting each patient’s painful days, and the results were assorted into three levels with 1 and 7 days as cut-off points [46]. The intensity of OP was based on the highest VAS_OP_ score over the seven days. Values were classified into three degrees, i.e., “absent”, “low”, and “high” with 1 and 10 units as cut-offs.

### 2.6. Statistical Analysis

Statistical analysis was conducted in IBM SPSS Statistics Version 25.0 [47] among the intention-to-treat population to decrease attrition bias [48]. VAS_DH_ scores were tested for this normality, and due to its non-normal distribution, Friedman and Kruskal–Wallis tests were conducted with Bonferroni corrections to compare the difference in the VAS_DH_ score among different timepoints and patients with varying OP perceptions, respectively. Mann–Whitney U tests were performed to compare the VAS_DH_ scores between teeth in LLLT and non-LLLT groups. Finally, a generalised estimating equation model (GEE) was established to investigate the effects of LLLT on DH for teeth with reduced and stable periodontium after confounders adjusted during orthodontic treatment and the first-year retention. The significance level was set as 0.05.

## 3. Results

### 3.1. Characteristics of Study Subjects

One hundred and forty-three teeth from 23 Chinese patients (2 males and 21 females) with DH fulfilling the eligibility criteria were included in this study and were randomly allocated to the LLLT group (n = 68) and non-LLLT group (n = 75) based on their attributes of opposing hemiarches. The two groups showed no significant difference in age, gender, tooth type, periodontal classification, perceptions of OP duration and intensity, and baseline VAS_DH_ score (*p* > 0.05) (Table 1). All 143 teeth that were initially randomized had at least three follow-ups (till T_3m_) and were included in the statistical analysis. The process of the randomized control trial is illustrated in the Consolidated Standards of Reporting Trial (CONSORT) flow diagram (Figure 2).

### 3.2. DH for Periodontally Compromised Patients Undergoing Orthodontic Treatment

Friedman tests for intragroup comparisons showed a generally decreasing tendency of VAS_DH_ over the fifteen timepoints for assessments including eight in active orthodontic treatment and seven in post-orthodontic retention (*p* < 0.001). During the active treatment stage, the VAS_DH_ score dropped rapidly during the first week of orthodontic treatment; then, the decreasing rate became slow, and the VAS_DH_ score reached its lowest at T_3m_. Subsequently, a very mild relapse was observed, with the peak noticeably lower than the baseline level. The VAS_DH_ score initially declined during the retention stage in the first week (T_deb_ to TR_1w_) and was maintained at this plateau level for a month (TR_1w_ to TR_1m_), followed by a slight increase thereafter (TR_3m_ to TR_12m_) (Figure 3).

### 3.3. DH in Periodontally Compromised Patients with Different Perceptions of OP Duration and Intensity

The results of comparisons among patients with different perceptions of OP duration showed a statistically significant difference in VAS_DH_ score at multiple timepoints. Before orthodontic treatment started, the VAS_DH_ score in patients with 1 to 7 days of OP was significantly higher than that in patients with over 7 days or less than 1 day of OP (*p* = 0.001). After orthodontic force loading, an immediate drop was observed in the VAS_DH_ score for patients with OP duration of less than 1 day and within 1 to 7 days, while the VAS_DH_ score for teeth in patients with more than 7 days of OP did not decrease until T_3m_ (Figure 4a). Specifically, VAS_DH_ scores for teeth in patients with OP perceptions of more than 7 days were significantly higher at T_1w_ compared to patients with OP duration of less than 1 day (*p* = 0.031). Moreover, teeth in patients with OP duration over 7 days had a higher VAS_DH_ score at T_1m_ than that in patients with OP duration less than 1 day (*p* = 0.020) and between 1 and 7 days (*p* = 0.012). During the retention stage, a significantly higher VAS_DH_ score was detected in patients with OP duration beyond 7 days compared to those with OP between 1 and 7 days at TR_3w_ (*p* = 0.013), TR_1m_ (*p* = 0.033), and TR_6m_ (*p* = 0.011), and compared to patients with OP less than 1 day at TR_3w_ (*p* = 0.044).

In terms of comparisons among patients with different perceptions of OP intensity, patterns of VAS_DH_ score variation were diverse (Figure 4b). The VAS_DH_ score for teeth in patients with high OP intensity was significantly higher than for those with low OP intensity at T_immd_ (*p* = 0.003) and TR_12m_ (*p* = 0.001), and higher than for those with absent OP experience at T_immd_ (*p* = 0.004), T_deb_ (*p* = 0.035), and TR_12m_ (*p* = 0.002). The VAS_DH_ score for teeth in patients with low OP intensity was significantly higher than for those with an absence of OP experience at T_3m_ (*p* < 0.001), T_deb_ (*p* = 0.007), and TR_6m_ (*p* = 0.001), and higher than patients with high OP intensity at TR_6m_ (*p* = 0.001) (Table 2).

### 3.4. Effects of LLLT on DH for Periodontally Compromised Patients Undergoing Orthodontic Treatment

Teeth in the LLLT group had a significantly lower VAS_DH_ score at T_12m_ for patients with an OP duration of less than 1 day (*p* = 0.032). The same condition was observed at T_3m_ (*p* = 0.023) and T_6m_ (*p* = 0.023) for patients with OP duration over 7 days. As for patients with different perceptions of OP intensity, LLLT presented a favourable outcome in DH alleviation at T_3m_ for patients with a low OP intensity (*p* = 0.034). No significant difference was found in DH relief for patients with an absence of OP or high OP intensity in all assessments (Table 3).

Generalized estimating equation modelling (GEE) was conducted to analyse the effects of LLLT on DH with other factors adjusted. Potential predictors included treatment timepoint, periodontal classification (stage and grade), tooth type, gender, age, and perception of OP (duration and intensity) (Table 4). The final model indicated that treatment timepoints, periodontal stages, patient’s age, and perceptions of OP (duration and intensity) were significant factors in the prediction of DH for teeth in periodontally compromised patients undergoing orthodontic treatment (*p* < 0.05). LLLT alone was not substantially effective (*p* = 0.376) on DH alleviation, but it had interaction effects with treatment timepoints (*p* < 0.001) (Table 5). Teeth in the LLLT group were estimated to have a significantly lower VAS_DH_ score (mean = 15.06; 95% CI: 10.55–21.49) than that for the non-LLLT group (mean = 21.89; 95% CI, 15.72–30.48) at the third month of orthodontic treatment (T_3m_, *p* = 0.011).

## 4. Discussion

This triple-blinded randomised controlled trial achieved the three objectives in terms of observing DH for teeth in periodontally compromised patients during orthodontic treatment and retention, analysing DH for teeth in patients with varying perceptions of OP, and evaluating the effects of LLLT on DH for teeth in periodontally compromised patients undergoing orthodontic treatment.

Our results showed that DH for teeth in periodontally compromised patients generally decreased throughout the orthodontic treatment, and the VAS_DH_ score was substantially lower in the retention stage than in the active treatment. Current research on the influence of orthodontic treatment on DH perceptions is insufficient and controversial. One rodent study found orthodontic tooth movement (OTM) could increase the trigeminal neurons’ excitability, thus making their receptive areas more sensitive to mechanical and thermal stimulations [49]. Sensory nerve fibres in the dental pulp are afferent endings of trigeminal neurons [50]; thus, a tooth might be more easily activated by external stimuli and experience DH. However, several clinical studies reported that pulp sensibility assessed by electric pulp testing (EPT) devices decreased significantly for teeth with OTM and showed a much higher threshold to respond but pulp sensibility did not alter substantially in reaction to thermal stimuli [51,52]. For the assessment of perceptions, evidence from animal studies is less solid than from clinical studies based on their indirect observations of subjects. Therefore, we assessed DH by patients’ judgement in response to a type of thermal stimulation (air blasts) during orthodontic treatment to obtain a relatively reliable result. The decreasing tendency of VAS_DH_ over the orthodontic treatment shown in our study was in line with the previous clinical studies using EPT readings [51,52,53] but inconsistent with their thermal test results [51,52]. One reason for this might be related to the fact that past studies adopted stronger cold stimulations on the tooth (delivered by cotton pellet dipped with refrigerant) and analysed its sensitivity based on positive/negative responses. Their approach is more feasible for the determination of pulp vitality but might omit some gentle and reversible changes in the dentine–pulp complex. We suspected the desensitising effects shown in our study and previous research could result from an escalation of the pulp sensibility threshold during orthodontic treatment, but further well-designed in vitro or in vivo studies are anticipated to confirm this postulation.

In addition, the results of our study demonstrated VAS_DH_ scores were significantly higher in patients with relatively high OP intensity and long OP duration than those with absent or brief OP experience in multiple assessments. DH for teeth is different in a periodontally compromised population with varying OP perceptions, indicating the two unfavourable perceptions have some relationship in the periodontal–orthodontic interdisciplinary context. Possible reasons might ascribe to the interactions of underlying mechanisms between DH and OP during orthodontic treatment. To begin with, OTM could induce immediate responses of both the periodontal ligament (PDL) and dental pulp in neural pathology and circulatory vasculature [54,55,56]. There is an increase in the release of neurotransmitters such as calcitonin gene-related peptide, substance P (SP), and endogenous opioids [57,58,59,60]. Later, with blood flow affected by OTM, various inflammatory cytokines were raised including but not limited to prostaglandins (PGs), interleukin (IL)-1β, IL-6, tumour necrosis factor-alpha (TNF-α), and interferon gamma (IFN-γ), and this may further change the PDL and dental pulp microenvironments [22]. Finally, nociceptors received by sensory nerve endings in both dental pulp and PDL are transmitted through the trigeminal ganglion, trigeminal nucleus, and thalamus into the brain somatosensory cortex [22,61]. The interpretations of this nociception could be adjusted by other factors on the individual level including the demographic domains (age, ethnicity, gender, etc.), psychosocial domains (anxiety, mood, satisfaction, etc.), and pathophysiologic domains (neuroendocrine-immune system) [62,63,64,65]. This study is the first to observe and verify the relationship between tooth DH and patients’ OP in the periodontal–orthodontic scenario.

The third objective of this study was to investigate the effects of LLLT on DH for teeth in periodontally compromised patients undergoing orthodontic treatment. Basic research on the mechanisms for the favourable effects of LLLT regarding DH alleviation was generalized in three aspects: first, LLLT alternates the neuronal physiology of sensory nerves and might contribute to immediate pain relief [66,67]; second, LLLT controls micro-inflammation within the dentine–pulp complex, which might reduce DH [68,69]; and third, LLLT could increase the viability of odontoblasts under its optimal settings and thus stimulate secondary dentine deposits [70,71,72]. As shown by our results, the VAS_DH_ score for teeth in the LLLT group was significantly lower than the non-LLLT group in periodontally compromised patients with OP duration of less than 1 day and more than 7 days. Previous research suggested OP initiation is related to physiological changes in sensory nerve endings and the release of neuropeptides, while OP persistence corresponds to the accumulation of inflammatory cytokines [22,57,58,59,60]. Taking account of the aforementioned mechanisms of LLLT on DH alleviation and potential biomedical interactions between DH and OP, it is possible that the LLLT’s desensitising effects in the periodontal–orthodontic scenario could be especially potent for patients with short and long OP durations in correspondence with neuronal physiological and inflammatory changes, respectively [68,69,73]. Further biomedical research is warranted to verify this interpretation. In terms of different perceptions of OP intensity, our results showed that the desensitising effects of LLLT on DH were significant for patients with low OP intensity, but no significant differences were detected for patients with absent or high OP intensities. This might be because the desensitising effects of LLLT on DH could be complexed by the combination of physiological and psychological factors concerning perceptions of OP intensity: for patients with absent OP perceptions, patients’ physiological pain thresholds are relatively high with trigeminal neurons hard to activate [66,67], while for patients who perceived themselves having high OP intensity, psychological factors such as catastrophizing might play the major role [74,75].

Based on the final GEE model with confounding factors adjusted, the desensitising effects of LLLT interact with orthodontic treatment progress. This suggests that LLLT has potential desensitising effects on DH for teeth in periodontally compromised patients undergoing orthodontic treatment; the efficacy was especially potent at the third month of active treatment. In addition, results of the final GEE model indicated that gender could impact DH in the periodontal–orthodontic context. To date, there is no consensus on gender differences towards DH: some research found women were more affected by DH than men [76,77], while others suggested males have a higher prevalence rate of DH [78,79]. The controversy among previous research is possibly related to the variations in the sampled populations, study designs, and confounding factors such as erosive dietary intakes and toothbrushing habits [80]. Therefore, although the final GEE model estimated a statistically significant higher VAS_DH_ score for male patients compared to females, it is worth bearing in mind that the number of male subjects included in this study was too limited (n = 26) to corroborate the gender difference towards DH.

In summary, this study is the first to observe DH for teeth in periodontally compromised patients undergoing orthodontic treatment. The findings address the lack of knowledge relating to the difference in DH among patients with varying OP perceptions and the effect of LLLT on DH in the periodontal–orthodontic context. There are some limitations of the study. First, unlike the previous clinical study that delivered LLLT to treated periodontal disease patients in short time intervals [81], we coordinated the laser application frequency with orthodontic adjustment needs for patients’ convenience [33]. Second, the observation period merely covered six months after laser irradiation, which was still relatively short to evaluate the persistent efficacy of LLLT [31]. Third, most patients who participated in this study were female adults due to their high motivation to improve aesthetics [82]. Although we adopted some statistical strategies such as using a split-mouth design and conducting GEE analysis to minimise potential bias that might be induced by gender imbalance, caution is required for result explication. Investigation into the long-term effects of LLLT on DH in male patients who undertake periodontal–orthodontic interdisciplinary treatment is still highly warranted.

## 5. Conclusions

DH for teeth in periodontally compromised patients generally decreases over the orthodontic treatment and retention period. The perception of DH is diverse in patients with different perceptions of OP. LLLT has potential desensitising effects on DH in the periodontal–orthodontic treatment context, and the efficacy is correlated with the progress of orthodontic treatment.

## Figures and Tables

**Figure 1 jcm-12-01419-f001:**
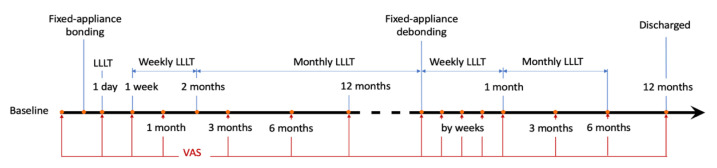
Orthodontic treatment and retention scheme with repeated LLLT for periodontally compromised patients (blue vertical lines indicate timepoints for interventions, red vertical lines indicate timepoints for assessments).

**Figure 2 jcm-12-01419-f002:**
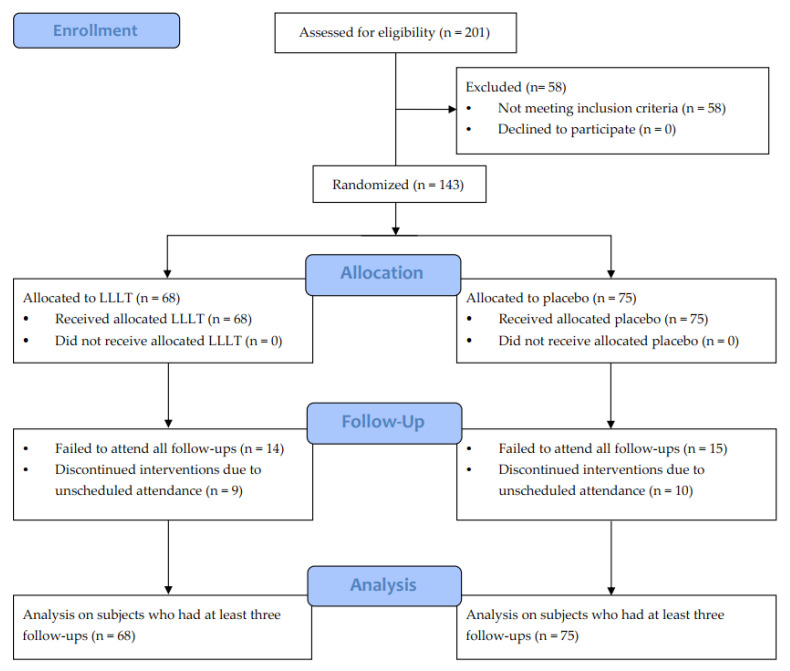
Consolidated Standards of Reporting Trial (CONSORT) flow diagram.

**Figure 3 jcm-12-01419-f003:**
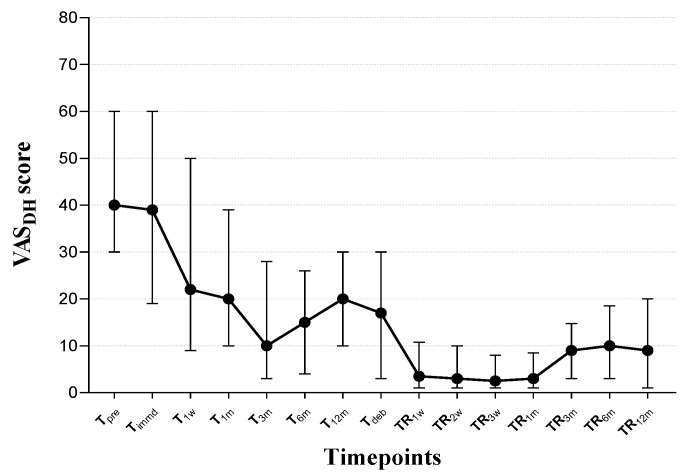
The plot describes the VAS_DH_ score (median and IQR) for teeth in periodontally compromised patients undergoing orthodontic treatment and retention.

**Figure 4 jcm-12-01419-f004:**
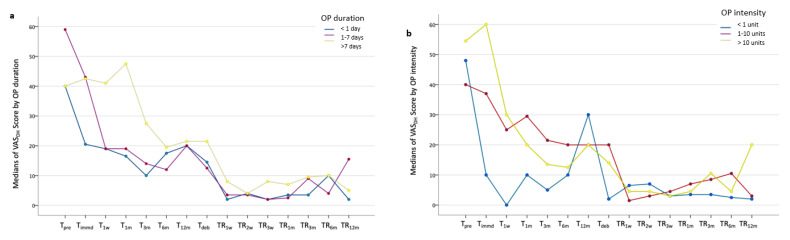
The line charts describe the VAS_DH_ score for teeth in periodontally compromised patients undergoing orthodontic treatment and retention with different perceptions of (**a**) OP duration and (**b**) OP intensity.

**Table 1 jcm-12-01419-t001:** Descriptive statistics for the teeth at the baseline (T_pre_).

			LLLT (n = 68)	Non-LLLT (n = 75)	Sig.
Age		Mean ± SD (median)	49.05 ± 8.03 (51)	48.21 ± 8.42 (50)	0.672
Gender		Male	9 (13.2%)	17 (22.7%)	0.144
		Female	59 (86.8%)	58 (77.3%)	
Tooth type		Incisors/canines	35 (51.5%)	44 (58.7%)	0.387
		Premolars	33 (48.5%)	31 (41.3%)	
Periodontal Classification	Stage	II	1 (1.5%)	1 (1.3%)	0.991
		III	22 (32.4%)	25 (33.3%)	
		IV	45 (66.2%)	49 (65.3%)	
	Grade	A	10 (14.7%)	14 (18.7%)	0.527
		B	58 (85.3%)	61 (81.3%)	
Orthodontic pain	Duration	<1 day	28 (41.2%)	30 (40%)	0.645
		1–7 days	20 (29.4%)	27 (36%)	
		>7 days	20 (29.4%)	18 (24%)	
	Intensity	<0 unit	18 (26.5%)	15 (20%)	0.602
		1–10 units	27 (39.7%)	30 (40%)	
		>10 units	23 (33.8%	30 (40%)	
VAS_DH_ score		Mean ± SD (median)	44.68 ± 17.99 (40)	47.77 ± 17.93 (41)	0.212

**Table 2 jcm-12-01419-t002:** VAS_DH_ scores for teeth in periodontally compromised patients with different perceptions of OP duration and OP intensity.

VAS_DH_ Score		Duration of OP				Intensity of OP (Peak VAS_OP_ Score)			
		<1 Day	1–7 Days	>7 Days	Sig.	Absent (<1 Unit)	Mild (1–10 Units)	High (>10 Units)	Sig.
		Med (IQR)	Med (IQR)	Med (IQR)		Med (IQR)	Med (IQR)	Med (IQR)	
**Active treatment stage**	**T_pre_**	40 (19.25)	59 (30) ^a,c^	40 (22.5)	0.001 *	48 (24)	40 (20)	54.5 (41.75)	0.096
	**T_immd_**	20.5 (32.5)	43 (32.5)	42.5 (38.5)	0.065	10 (63)	37 (27.25)	60 (30) ^d,e^	0.001 *
	**T_1w_**	19 (52.5)	19 (43)	41 (29.25) ^a^	0.035 *	0 (70)	25 (43)	30 (34)	0.256
	**T_1m_**	16.5 (22.75)	19 (12)	47.5 (40.25) ^a,b^	0.008 *	10 (26)	29.5 (46)	20 (13.5)	0.128
	**T_3m_**	10 (24.75)	14 (20)	27.5 (29.5)	0.108	5 (10)	21.5 (23.5) ^d^	13.5 (25.25)	0.001 *
	**T_6m_**	17.5 (18)	12 (19)	19.5 (23.75)	0.353	10 (26)	20 (18.75)	12.5 (17.75)	0.327
	**T_12m_**	20 (18.75)	20 (17.25)	21.5 (25.5)	0.789	30 (45.75)	20 (15.5)	20 (17)	0.612
	**T_deb_**	14.5 (22.75)	12.5 (15)	21.5 (30.75)	0.111	2 (20)	20 (22.25) ^d^	14 (19) ^d^	0.008 *
**Retention stage**	**TR_1w_**	2 (14.5)	3.5 (9)	8 (22.5)	0.460	6.5 (18)	1.5 (9.75)	4.5 (9.5)	0.289
	**TR_2w_**	4 (19.5)	3.5 (5.25)	4 (13)	0.415	7 (28.25)	3 (9.25)	4.5 (9)	0.467
	**TR_3w_**	2 (11.5)	2 (2.25)	8 (11.5) ^a,b^	0.007 *	3 (18.5)	4.5 (8.5)	3 (2.75)	0.855
	**TR_1m_**	3.5 (14.75)	2.5 (5.5)	7 (8) ^b^	0.039 *	3.5 (14.75)	7 (8)	4.5 (7.25)	0.806
	**TR_3m_**	3.5 (11.75)	9 (11.5)	9.5 (12)	0.423	3.5 (11.75)	8.5 (9.25)	10.5 (12.5)	0.283
	**TR_6m_**	10 (19.75)	4 (4.5)	10 (6) ^b^	0.010 *	2.5 (3.5)	10.5 (16.25) ^d,f^	4.5 (6.5)	0.000 *
	**TR_12m_**	2 (17.5)	15.5 (18.25)	5 (19)	0.047	2 (10)	3 (10.25)	20 (17.5) ^d,e^	0.000 *
**Intragroup comparison**		<0.001 *	<0.001 *	<0.001 *		<0.001 *	<0.001 *	<0.001 *	

* *p* < 0.05, Kruskal–Wallis test and Friedman test for intergroup and intragroup comparisons, respectively, followed by Bonferroni post hoc adjustment; ^a^ significantly higher than the VAS_DH_ score for teeth in patients with OP less than one day; ^b^ significantly higher than the VAS_DH_ score for teeth in patients with OP between 1 day and 7 days; ^c^ significantly higher than the VAS_DH_ score for teeth in patients with OP more than 7 days; ^d^ significantly higher than the VAS_DH_ score for teeth in patients with a peak VAS_OP_ score of less than 1 unit; ^e^ significantly higher than the VASDH score for teeth in patients with a peak VASOP between 1 day and 10 units; ^f^ significantly higher than the VASDH score for teeth in patients with a peak VASOP of more than 10 units.

**Table 3 jcm-12-01419-t003:** Intragroup and intergroup comparisons of VAS_DH_ scores for teeth in periodontally compromised patients undergoing orthodontic treatment.

VAS_DH_ Score		Duration of OP									Intensity of OP (Peak VAS_OP_ Score)								
		<1 Day			1–7 Days			>7 Days			<1 Unit			1–10 Units			>10 Units		
		LLLT	Non-LLLT	Sig	LLLT	Non-LLLT	Sig	LLLT	Non-LLLT	Sig	LLLT	Non-LLLT	Sig	LLLT	Non-LLLT	Sig	LLLT	Non-LLLT	Sig
		Med (IQR)	Med (IQR)		Med (IQR)	Med (IQR)		Med (IQR)	Med (IQR)		Med (IQR)	Med (IQR)		Med (IQR)	Med (IQR)		Med (IQR)	Med (IQR)	
**Active treatment stage**	**T_pre_**	30 (10)	40 (19)	0.062	56 (35)	59 (30)	0.651	43 (29)	40 (23)	0.497	39.5 (10)	48 (24)	0.111	36.5 (27)	40 (20)	0.537	50 (37)	54.5 (42)	0.949
	**T_immd_**	30 (40)	20.5 (33)	0.579	44.5 (53)	43 (33)	0.683	22 (50)	42.5 (39)	0.292	20 (64)	10 (63)	0.565	30 (40)	37 (27)	0.773	44.5 (61)	60 (30)	0.182
	**T_1w_**	19.5 (25)	19 (53)	0.913	37.5 (47)	19 (43)	0.277	24.5 (37)	41 (29)	0.095	10 (19)	0 (70)	0.724	25 (39)	25 (43)	0.835	32.5 (41)	30 (34)	0.808
	**T_1m_**	10.5 (39)	16.5 (23)	0.448	20 (11)	19 (12)	0.763	11.5 (54)	47.5 (40)	0.131	8.5 (20)	10 (26)	0.197	28.5 (51)	29.5 (46)	1.000	19 (13)	20 (14)	0.424
	**T_3m_**	8 (10)	10 (25)	0.080	8 (32)	14 (20)	0.706	3.5 (18)	27.5 (30)	0.023 *	8 (10)	5 (10)	0.985	10 (29)	21.5 (24)	0.034 *	4 (32)	13.5 (25)	0.091
	**T_6m_**	12 (19)	17.5 (18)	0.605	20 (35)	12 (19)	0.518	6 (20)	19.5 (24)	0.023 *	10 (23)	10 (26)	0.687	15 (27)	20 (19)	0.209	11 (31)	12.5 (18)	0.615
	**T_12m_**	9 (24)	20 (19)	0.032 *	13 (22)	20 (17)	0.300	20 (36)	21.5 (26)	0.965	9 (30)	30 (46)	0.096	17 (29)	20 (16)	0.560	12.5 (20)	20 (17)	0.339
	**T_deb_**	18 (30)	14.5 (23)	0.751	17.5 (19)	12.5 (15)	0.780	15.5 (34)	21.5 (31)	0.519	0 (25)	2 (20)	1.000	28 (28)	20 (22)	0.236	11 (18)	14 (19)	0.138
**Retention stage**	**TR_1w_**	10 (20)	2 (15)	0.747	2 (10)	3.5 (9)	0.986	3 (4)	8 (23)	0.436	10 (15)	6.5 (18)	0.828	3 (11)	1.5 (10)	0.549	5 (8)	4.5 (10)	0.592
	**TR_2w_**	3 (18)	4 (20)	0.684	4 (8)	3.5 (5)	0.590	3 (9)	4 (13)	0.545	6.5 (13)	7 (28)	0.592	3 (15)	3 (9)	0.620	2 (8)	4.5 (9)	0.601
	**TR_3w_**	0 (6)	2 (12)	0.381	2 (5)	2 (2)	0.661	2 (11)	8 (12)	0.148	0 (2)	3 (19)	0.050	4 (13)	4.5 (9)	0.921	2 (3)	3 (3)	0.482
	**TR_1m_**	1 (8)	3.5 (15)	0.267	2 (7)	2.5 (6)	0.930	3 (5)	7 (8)	0.085	1 (11)	3.5 (15)	0.330	3 (5)	7 (8)	0.271	2 (7)	4.5 (7)	0.357
	**TR_3m_**	3 (16)	3.5 (12)	0.781	9 (24)	9 (12)	0.569	9 (10)	9.5 (12)	0.606	7 (17)	3.5 (12)	0.829	7 (10)	8.5 (9)	0.692	10.5 (24)	10.5 (13)	0.655
	**TR_6m_**	10 (14)	10 (20)	0.809	5.5 (14)	4 (5)	0.427	20 (21)	10 (6)	0.112	6.5 (13)	2.5 (4)	0.274	18 (20)	10.5 (16)	0.345	4 (13)	4.5 (7)	0.921
	**TR_12m_**	2 (10)	2 (18)	0.370	12 (14)	15.5 (18)	0.736	9.5 (27)	5 (19)	0.703	2 (7)	2 (10)	0.693	10 (18)	3 (10)	0.224	19 (19)	20 (18)	0.362
**Intragroup comparison**		<0.001 *	<0.001 *		<0.001 *	<0.001 *		<0.001 *	<0.001*		<0.001 *	<0.001 *		<0.001 *	<0.001 *		<0.001 *	<0.001 *	

* Mann–Whitney U test for intergroup comparisons; Friedman test for intragroup comparisons.

**Table 4 jcm-12-01419-t004:** Complete and final generalised estimating equation (GEE) models for the VAS_DH_ score.

Parameters	Complete Model			Final Model		
	𝜒^2^	df	Sig.	𝜒^2^	df	Sig.
**(Intercept)**	138.358	1	0.000 *	467.840	1	0.000 *
**LLLT**	0.185	1	0.414	0.784	1	0.376
**Timepoints**	3579.396	14	0.000 *	2178.586	14	0.000 *
**Periodontal stage**	91.927	2	0.000 *	77.735	2	0.000 *
**Periodontal grade**	12.457	1	0.000 *	-	-	-
**OP duration**	6.016	2	0.049 *	6.510	2	0.039 *
**OP intensity**	2.557	2	0.279	12.878	2	0.002 *
**Tooth type**	2.327	1	0.127	-	-	-
**Gender**	0.163	1	0.687	26.294	1	0.000 *
**Age**	12.271	1	0.000 *	-	-	-
**timepoints × ** **LLLT**	381.866	14	0.000 *	154.167	14	0.000 *

* *p* < 0.05.

**Table 5 jcm-12-01419-t005:** VAS_DH_ estimates for teeth in periodontally compromised patients undergoing orthodontic treatment in the two groups based on the final GEE model.

VAS_DH_ Estimates		LLLT			Non-LLLT			Sig.
		Mean	95% CI		Mean	95% CI		
			Lower	Upper		Lower	Upper	
**Active treatment stage**	**T_pre_**	57.57	43.12	76.86	60.29	45.52	79.85	0.682
	**T_immd_**	45.65	31.41	66.34	47.27	33.52	66.68	0.692
	**T_1w_**	35.97	25.05	51.65	37.10	23.71	58.04	0.839
	**T_1m_**	28.27	19.30	41.41	32.26	21.90	47.54	0.291
	**T_3m_**	15.06	10.55	21.49	21.89	15.72	30.48	0.011 *
	**T_6m_**	19.94	14.01	28.38	23.27	14.51	37.31	0.453
	**T_12m_**	22.57	15.17	33.56	27.27	18.36	40.49	0.409
	**T_deb_**	19.93	14.64	27.13	19.18	13.02	28.25	0.812
**Retention stage**	**TR_1w_**	8.99	5.93	13.62	9.09	5.15	16.06	0.949
	**TR_2w_**	8.67	5.80	12.94	8.05	4.30	15.05	0.765
	**TR_3w_**	5.59	3.91	7.98	6.89	3.99	11.89	0.444
	**TR_1m_**	5.32	3.87	7.31	6.87	4.38	10.78	0.147
	**TR_3m_**	13.10	8.13	21.12	12.70	8.41	19.17	0.835
	**TR_6m_**	12.99	8.75	19.31	11.43	6.61	19.77	0.605
	**TR_12m_**	12.97	8.27	20.35	14.43	8.57	24.32	0.653

* *p* < 0.05.

## Data Availability

The data that support the findings of this study are available from the corresponding author, Y.Y., upon reasonable request.

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
