# Peer review of "Effects of Low-Level Laser Therapy on Dentin Hypersensitivity in Periodontally Compromised Patients Undergoing Orthodontic Treatment: A Randomised Controlled Trial"

_jcm, 2023, doi:10.3390/jcm12041419_

Round 1

Reviewer 1 Report

The effects of low-level laser therapy on dentin hypersensitivity in  patients undergoing orthodontic treatment is a very important topic nowadays.

The study is well done, I am pleased to have reviewed this paper. 

The study aimed to assess the effects of low-level laser therapy on dentin hypersensitivity in periodontally compromised patients undergoing orthodontic treatment. The hypothesis is explained and a randomized-controlled trial has been performed. 

Keywords are appropiate, the methodology is sound.

The results and discussion chapters are well written. 

Figure 1: Consolidated Standards of Reporting Trial (CONSORT) flow diagram is clear. 

Figure 3 describes the VASDH score (median and IQR) for teeth in periodontally compromised patients undergoing orthodontic treatment and retention and is of high quality. 

Intragroup and intergroup comparisons of VASDH scores for teeth in periodontally patients undergoing orthodontic treatment are presented and adequate statistical analysis has been performed. 

The conclusion is sustained by the results. 

Author Response

We sincerely appreciate the reviewer’s comments. 

Reviewer 2 Report

Summary of the manuscript

This manuscript based on the clinical trial on Effects of low-level laser therapy on dentin hypersensitivity in periodontally compromised patients undergoing orthodontic treatment: A randomized-controlled trial involved 143 teeth with the obejctives: 1) observing DH for teeth in periodontally compromised patients during orthodontic treatment and retention, 2) analyzing DH in periodontally compromised patients with varying perceptions of OP, and 3) evaluating the effects of LLLT on DH for teeth in periodontally compromised patients undergoing orthodontic treatment.

Revision requires as below:

1.      Tittle- clear and reflec the research project.

2.      Abstract: Statistical analysis and significant of p value  can be added.

3.      Introductions: Page 2 line 72, remove “by our group”, line 80 remove ‘we’ and change to this study.

4.      Material and method:

-Recruitment of the sample- inclusion and exclusion criteria may need to include the type of malocclusion and the severity of the malocclusions eg degree of the crowding and its all bracket and be engage at the same time because it will give different effects to the patients.

-the orthodontic procedure only explain until the aligning stage, starting with 014 niti and 016 niti archwire….what happen after that before the patient can be debonded because the researches assess the VAS up to post debonding according to figure 2. The authors should explain the procedure of the fixed appliances in details up to debonding procedure in the methods part because it can affect the data collections

- study design: page 3 line 127-129 , the sentences can be removed from this part and should be added in the result section because that is the part of the data and results.

5.      Results:ok

6.      Discussion: sample size only involve 2 males and 21 females, may need to discuss in the discussion the effects of the gender towards the pain perceptions. its any bias towards the currents results.

7.      The references: Please read and follow the journal guidelines.

 Author Response

Response to the reviewer:

Q1. Title- clear and reflect the research project.

A1. Thank you for the comments.

Q2. Abstract: Statistical analysis and significant of p value can be added.

A2. Thank you for the comments. Statistical analysis and significant of p-value have been added in the abstract. In detail,

1). For DH across the whole treatment process, the p-value is less than 0.001 by Friedman test (Page 1 line 17 and 19).

2). According to Kruskal-Wallis tests, DH in patients with different OP duration at Tpre, T1w, T1m, TR3w, TR1m, and TR6m and with different OP intensity at Timmd, T3m, Tdeb, TR6m, and TR12m presented significant difference (Page 1 line 18). Since every timepoint has a p-value, we generally describe them in the abstract part as p < 0.05 (Page 1 line 20).

3). Results of the GEE model suggested a significant difference in DH between LLLT and non-LLLT groups at the T3m, the exact p-value (p = 0.011) has been listed to make it clear. (Page 1 line 22)

Q3. Introductions: Page 2 line 72, remove “by our group”, line 80 remove ‘we’ and change to this study.

A3: We thank the reviewer for the explicit advice and have made the revisions accordingly.

Q4. Material and method:

A4. The suggestions given by the reviewer are very insightful. We have revised the “materials and methods” section. In detail,

-Recruitment of the sample- inclusion and exclusion criteria may need to include the type of malocclusion and the severity of the malocclusions eg degree of the crowding and its all bracket and be engaged at the same time because it will give different effects to the patients.

According to previous research, the severity of malocclusion and degree of crowding has no effect on perceived discomfort experienced by patients undergoing orthodontic treatment [1, 2]. Besides, this does not directly influent the treatment modalities to be taken. Therefore, we didn’t use these factors to define our eligibility criteria. Instead, we included teeth with reduced periodontium in patients who undertook adjunctive orthodontic treatment [3], acting as a part of occlusal therapy to facilitate the equilibrium of forces and eliminate occlusal interference, and excluded teeth in patients who undertook comprehensive orthodontic treatment [4, 5]. In the revised manuscript, we have clarified the eligibility criteria (Page 3 line 95 to 118), and also discussed the reasons (Page 3 line 96-100 and 114-116).

 -the orthodontic procedure- only explain until the aligning stage, starting with 014 niti and 016 niti archwire….what happen after that before the patient can be debonded because the researches assess the VAS up to post debonding according to figure 2. The authors should explain the procedure of the fixed appliances in details up to debonding procedure in the methods part because it can affect the data collections

1). We expanded the whole treatment procedure until debonding in detail (Page 4 line 146 to 153).

2). As for reasons of our treatment procedures, first, the pain investigations mainly focus on the first three months (with orthodontic pain evaluated by a 7-day pain dairy immediately after the fixed appliances were bonded) [6]. Second, one of the objectives of this study is to investigate the relationship between OP and DH. Therefore, we adopted the same archwire protocol in the initial stage of treatment (0.014 NiTi for two months and 0.016 NiTi for one month), which is supported by other studies on orthodontic pain [1,7]. Third, following the 0.016 NiTi archwire, the same operator sequentially prescribed larger and stiffer archwires depending on patients’ individual conditions, which was an approach for standardization. Due to ethical concerns, we didn’t prolong the treatment period to make all procedures identical, and the patients were debonded after they had achieved their own treatment objectives (a lot of patients were debonded shortly after one year). Since the active treatment periods vary, we didn’t evaluate tooth DH after T12m until fixed appliances were debonded (Tdeb), as indicated in Figure 2. In the revised manuscript, we clarified the orthodontic treatment with reasons explained in detail (Page 4 line 141 to 157).

 - study design- page 3 line 127-129, the sentences can be removed from this part and should be added in the result section because that is the part of the data and results.

Thank you for the comment. The sentences on page 3 line 127-129 have been moved to the “Result” section (Page 6 lines 211 to 214).

Q5. Results: ok

A5. We thank the reviewer for the comments.

Q6. Discussion: sample size only involve 2 males and 21 females, may need to discuss in the discussion the effects of the gender towards the pain perceptions. its any bias towards the currents results.

A6. We agree with the reviewer that gender imbalance is one of the limitations of this study, which was added on Page 12 line 400 to 406. Besides, we expanded the discussion on gender difference towards DH (Page 11 line 382 to 391).

Q7. The references: Please read and follow the journal guidelines.

A7. We thank the reviewer for the kind reminder. All references have been carefully checked and revised following the journal guidelines. References below which are highlighted (yellow) in the manuscript are newly added. (All the references are shown as revised in track records as the reference sequence numbers are changed after new references are added.)

References of the response to reviewers

  1. Abdelrahman, R.S.; Al-Nimri, K.S.; Al Maaitah, E.F. Pain experience during initial alignment with three types of nickel-titanium archwires: A prospective clinical trial. The Angle Orthodontist 2015, 85, 1021-1026, doi:10.2319/071614-498.1.
  2. Krishnan, V. Orthodontic pain: from causes to management--a review. The European Journal of Orthodontics 2007, 29, 170-179, doi:10.1093/ejo/cjl081.
  3. Rabie, A.B.; Deng, Y.; Jin, L.J. Adjunctive orthodontic treatment of periodontally involved teeth: case reports. Quintessence Int 1998, 29, 13-19.
  4. Leiva Villagra, N.; Muñoz Domon, M.; Véliz Méndez, S. Comprehensive orthodontic treatment of adult patient with cleft lip and palate. Case Rep Dent 2014, 2014, 795342, doi:10.1155/2014/795342.
  5. Deng, Y.; Sun, Y.; Xu, T. Evaluation of root resorption after comprehensive orthodontic treatment using cone beam computed tomography (CBCT): a meta-analysis. BMC Oral Health 2018, 18, 116, doi:10.1186/s12903-018-0579-2.
  6. Long, H.; Wang, Y.; Jian, F.; Liao, L.-N.; Yang, X.; Lai, W.-L. Current advances in orthodontic pain. International Journal of Oral Science 2016, 8, 67-75, doi:10.1038/ijos.2016.24.
  7. Farzanegan, F.; Zebarjad, S. M.; Alizadeh, S.; Ahrari, F. Pain reduction after initial archwire placement in orthodontic patients: A randomized clinical trial. American Journal of Orthodontics and Dentofacial Orthopedics 2012, 141, 169–173. 

Round 2

Reviewer 2 Report

comments:

1. inclusion criteria: the author wrote "The severity of malocclusion and degree of crowding does not affect the perceived discomfort experienced by patients undergoing orthodontic treatment [35,36]. Therefore, we didn’t list these factors in our eligibility criteria"

the severity of the crowding is require to be considered in the inclusion criteria because its reflex the wire enggament of during treatment and will affect the respons from patients regarding VASDH scores. Thus the severity of the malocclusion is require to be put in as inclusion criteria.

2. Authors still did not present a full sample size calculation which conclude the final sample require in this study.

3."Subsequently, round/rectangular NiTi archwires with larger sizes and stainless steel archwires (G&H Orthodontics, Franklin, IN, USA) were sequentially delivered to the patients depending on individual circumstances

-wire size should be mentions before its reflex the methods of the trials.Please revise this part again.

Author Response

We thank the reviewer for offering constructive comments. We’ve made revisions to the following three aspects:

Comment 1:

inclusion criteria: the author wrote "The severity of malocclusion and degree of crowding does not affect the perceived discomfort experienced by patients undergoing orthodontic treatment [35,36]. Therefore, we didn’t list these factors in our eligibility criteria"

the severity of the crowding is require to be considered in the inclusion criteria because its reflex the wire engagement of during treatment and will affect the respons from patients regarding VASDH scores. Thus the severity of the malocclusion is require to be put in as inclusion criteria.

Response 1:

We agree with the reviewer that the severity of the crowding is related to wire engagement during treatment. When recruiting the patient, we evaluated the severity of contact point displacement based on IOTN-DHC. The reason for not using space analysis to evaluate the severity of crowding is that the majority of the patients we included in this study had more than five teeth loss due to periodontitis (periodontal stage IV) and some cases had first permanent molars lost. Only patients whose maximum contact point was higher than 4 mm (scored 4d) were included in this study. We revised the manuscript by adding this in the inclusive criteria on page 3 under the 2.1.1 section (shown in red).

Comment 2:

Authors still did not present a full sample size calculation which conclude the final sample require in this study.

Response 2:

The sample size calculation is described on page 3 under the 2.2 section (in red). The sample size was based on a previous study with similar LLLT settings. To detect a difference in DH between LLLT and non-LLLT groups, 82 teeth were calculated by the software G*Power at the significant level of 5% to reach 80% power. Considering 15% of dropout rate, we need to include at least 123 teeth at the baseline.

Among the 201 screened teeth, 58 teeth didn’t meet the inclusion criteria and we finally enrolled 143 teeth for treatment. The analysis was conducted among all 143 intention-to-treat teeth which all had data from at least three follow-ups (till T3m). We have revised the manuscript on page 5 section 3.1 (shown in red) and figure 2 CONSORT diagram to have this issue clarified.

Comment 3:

"Subsequently, round/rectangular NiTi archwires with larger sizes and stainless steel archwires (G&H Orthodontics, Franklin, IN, USA) were sequentially delivered to the patients depending on individual circumstances

-wire size should be mentions before its reflex the methods of the trials. Please revise this part again.

Response 3:

Thank you for the comments. We revised the manuscript by documenting the archwire sizes in detail, which were 0.018-inch, 0.017 × 0.025-inch NiTi archwires and 0.017 × 0.025-inch stainless steel archwires (Page 4 Section 2.4.1, shown in red). The archwires were sequentially prescribed based on individual conditions for the following visits.